# Molecular Mechanisms for the Vicious Cycle between Insulin Resistance and the Inflammatory Response in Obesity

**DOI:** 10.3390/ijms24129818

**Published:** 2023-06-06

**Authors:** Dariusz Szukiewicz

**Affiliations:** Department of Biophysics, Physiology & Pathophysiology, Faculty of Health Sciences, Medical University of Warsaw, 02-004 Warsaw, Poland; dariusz.szukiewicz@wum.edu.pl

**Keywords:** insulin resistance, insulin sensitivity, innate immune system, adaptive immune system, inflammatory response, obesity, visceral adipose tissue, insulin, insulin receptor, insulin signaling

## Abstract

The comprehensive anabolic effects of insulin throughout the body, in addition to the control of glycemia, include ensuring lipid homeostasis and anti-inflammatory modulation, especially in adipose tissue (AT). The prevalence of obesity, defined as a body mass index (BMI) ≥ 30 kg/m^2^, has been increasing worldwide on a pandemic scale with accompanying syndemic health problems, including glucose intolerance, insulin resistance (IR), and diabetes. Impaired tissue sensitivity to insulin or IR paradoxically leads to diseases with an inflammatory component despite hyperinsulinemia. Therefore, an excess of visceral AT in obesity initiates chronic low-grade inflammatory conditions that interfere with insulin signaling via insulin receptors (INSRs). Moreover, in response to IR, hyperglycemia itself stimulates a primarily defensive inflammatory response associated with the subsequent release of numerous inflammatory cytokines and a real threat of organ function deterioration. In this review, all components of this vicious cycle are characterized with particular emphasis on the interplay between insulin signaling and both the innate and adaptive immune responses related to obesity. Increased visceral AT accumulation in obesity should be considered the main environmental factor responsible for the disruption in the epigenetic regulatory mechanisms in the immune system, resulting in autoimmunity and inflammation.

## 1. Introductory Overview

The Nobel Prize in Physiology or Medicine 1923 was awarded jointly to Frederick Grant Banting and John James Rickard Macleod “for the discovery of insulin”. One hundred years later, knowledge of how insulin works is not just about blood glucose regulation. The comprehensive metabolic effects of this anabolic hormone throughout the body also include lipid and protein homeostasis [1]. Particularly noteworthy is the anti-inflammatory effect of insulin, the only glucose-lowering hormone in the body, manifested by the modulation of inflammatory mediators and direct action upon immune cells to enhance immunocompetence [2]. Impaired tissue sensitivity to insulin or insulin resistance (IR) paradoxically leads to diseases with an inflammatory component despite hyperinsulinemia [3]. The prevalence of obesity, defined as a body mass index (BMI) ≥ 30 kg/m^2^, has been increasing worldwide on a pandemic scale with accompanying syndemic health problems, including IR and diabetes [4]. Moreover, in response to IR, hyperglycemia itself stimulates a primarily defensive inflammatory response associated with the subsequent release of numerous inflammatory cytokines and a real threat of organ function deterioration [5]. This is so-called low-grade inflammation, usually defined as “the chronic production, but a low-grade state, of inflammatory factors”. In addition to hyperglycemia, the conditions characterized by low-grade inflammation include obesity, depression, and chronic pain, among others [6,7,8].

### 1.1. Insulin

Insulin was the first peptide hormone discovered in 1922 during the examination of pancreatic extracts in diabetes research [9,10]. However, it took nearly two and a half decades to achieve the physicochemical characterization of insulin [11]. Finally, after determining the amino acid sequence of insulin using Sanger sequencing in 1955, it became clear that, as a heterodimer composed of 51 amino acids, the insulin molecule consists of two chains: a 21-residue A-chain linked to a 30-residue B chain by two disulfide bonds derived from cysteine residues (A7–B7 and A20–B19). An intrachain disulfide bond also exists within the A-chain (A6–A11) [12] (Figure 1). Thus, insulin produced and secreted by β-cells in the pancreatic islets of Langerhans and encoded in humans by the INS gene became the first protein to have its sequence determined. Interestingly, in contrast to the long-held belief that insulin production is unique to the pancreas, it was demonstrated that the human fetus can spontaneously produce insulin outside the pancreas in a subset of fetal enteroendocrine K/L cells in the small intestine [13]. Insulin production can also be induced in intestinal epithelial cells during self-adaptation against a diabetic environment [14]. Extra-pancreatic synthesis of insulin in male germ line cells and human endometrium may play a crucial role in the regulation of reproductive processes, especially in the support of spermatozoa during their flow toward the site of fertilization [15]. In the central nervous system, insulin is synthesized by a subpopulation of neurons in the cerebral cortex and neural progenitor cells in the hippocampus [16]. The molecular formula for human insulin is C257H383N65O77S6, and mature human insulin has a molecular mass of 5808 Da [17]. Although this hormone is a small protein, it contains almost all the structural features typical of proteins, including α-helix, β-sheet, β-turn, high-order assembly, allosteric T (tense)- and R (relaxed)-transition, and conformational changes in amyloidal fibrillation [18]. The structure of insulin contains determinants of foldability, trafficking, self-assembly, and receptor binding [19,20].

Insulin is part of a larger superfamily of evolutionary sequence- and fold-related active proteins (hormones), which have been found not only in mammals, birds, reptiles, amphibians, fish, and cephalochordates but also in Mollusca, insects, and Caenorhabditis elegans. In addition to insulin, this insulin-like superfamily includes, among others, the relaxin peptides relaxin-1, -2, and -3; the insulin-like growth factors (IGF-1 and IGF-2); the insulin-like peptides (INSL3—mammalian Leydig cell-specific insulin-like peptide, INSL4—early placenta insulin-like peptide (EPIL), INSL5, INSL6, and Caenorhabditis elegans insulin-like peptides); the locust insulin-related peptide (LIRP); molluscan insulin-related peptides (MIPs); and the insect prothoracicotropic hormone (bombyxin) [21,22,23,24,25,26,27]. The sequence homology for the members within the human insulin–relaxin superfamily is presented in Figure 2.

All members of the superfamily exhibit the same disulfide-bonding pattern that was determined originally for human insulin. This means that the fold in the protein molecule comprises two polypeptide chains (A and B) linked by two disulfide bonds: all share a conserved arrangement of four cysteines in their A chain, the first of which is linked by a disulfide bond to the third, while the second and fourth are linked by interchain disulfide bonds to cysteines in the B chain [29].

#### Insulin Synthesis and Posttranslational Modification of Its Tertiary Structure (Figure 2)

Almost exclusively produced in pancreatic β-cells, insulin is the biosynthetic end-product of the enzymatic decomposition of its single-chain precursor, known as preproinsulin, which is the initial result of the process of insulin mRNA translation. In a preproinsulin molecule, the following are present sequentially from the N-terminus: the signal peptide, insulin B-chain, C-peptide, and insulin A-chain. The signal peptide drives the nascent polypeptide of insulin from the cytosol into the lumen of the endoplasmic reticulum (ER), the starting point of the secretory pathway [30]. Thus, the proteolytic processing of preproinsulin is coupled to trafficking between cellular compartments. Cotranslational and post-translational translocation of preproinsulin into the ER is due to the presence of the translocation-associated protein complex (TRAP), also called signal sequence receptor (SSR), in the ER. The TRAP consists of four integral membrane proteins, namely, TRAPα/SSR1, TRAPβ/SSR2, and TRAPδ/SSR4, with their extramembranous portion directed primarily into the ER lumen, whereas the extramembranous portion of TRAPγ/SSR3 is mainly cytosolic [31]. At the level of the ER, the signal peptide is cleaved from preproinsulin by the signal peptidase, allowing the molecule to be converted to proinsulin. Proinsulin folds on the luminal side of the ER, forming three evolutionarily conserved disulfide bonds. Next, folded proinsulin in the form of dimers exits from the ER and traffics through the Golgi complex into immature secretory granules. After excision of the C-peptide by a specialized set of endoproteases and carboxypeptidase activity, fully bioactive two-chain insulin ready for immediate secretion is ultimately stored as microcrystalline arrays of zinc insulin hexamers in mature glucose-regulated granules [20]. Defects in the pancreatic β-cell secretion system are reported in conditions linked to insulin resistance, such as prediabetes and type 2 diabetes (T2D), and include impaired proinsulin processing coexisting with a deficit in mature insulin-containing secretory granules [32].

C-peptide and insulin enter the bloodstream at the same time and in equal amounts. However, in contrast to insulin, C-peptide is not extracted by the liver or other organs; therefore, it has a longer half-life of approximately 30 min compared with only 6 min for insulin. Thus, C-peptide reflects endogenous insulin secretion more accurately than insulin and—for that reason—is widely used as a clinical marker for pancreatic β-cell function [32,33]. Moreover, many years of research on both humans and animals have shown that C-peptide is much more than a byproduct of insulin biosynthesis [34,35,36,37]. Although it does not participate in the regulation of glucose levels, C-peptide might play a role in preventing and potentially reversing some of the chronic complications of diabetes, including vascular and nervous damage [38,39]. It remains an open question whether these activities related to the therapeutic potential of C-peptide are mediated by the receptor, the existence of which is still uncertain, or whether an alternative nonreceptor-mediated mechanism is involved [40,41].

### 1.2. Structure of the Insulin Receptor (INSR)

Because insulin or insulin signaling through the INSR plays a key role in the regulation of glucose homeostasis, enabling body cells to access glucose in the blood, virtually all cells express the INSR [42,43]. The action of this cell-surface receptor thus translates into whole-body nutrient homeostasis, and INSR malfunction in humans is linked to various disorders, such as type 2 diabetes mellitus (T2DM), obesity, metabolic syndrome, polycystic ovary syndrome (PCOS), atherosclerotic cardiovascular diseases, neurodegenerative disorders including Alzheimer’s disease (AD), and multiple cancers [44,45,46,47,48,49,50,51] (see also: Section 2. Insulin Resistance).

The INSR’s endogenous ligands include insulin, IGF-1, and IGF-2. Similar to the receptors for other protein hormones, the INSR belongs to the large class of receptor tyrosine kinases and is located transmembraneously in the plasma membrane of insulin-sensitive cells [52]. The INSR is a heterotetrameric protein composed of two entirely extracellular α subunits (MW 135 kDa) and two penetrating intracellular β subunits (MW 95 kDa). These subunits are covalently linked by disulfide bonds that determine their α2β2 tetrameric structure. The α-chains of the INSR house insulin-binding domains that constitute most of the INSR ectodomains, whereas the β-chains constitute the transmembrane and intracellular (tyrosine kinase) domains of the INSR [53,54,55]. The receptor is encoded by a single gene that is located on human chromosome 19 and consists of 22 exons (exons 1–11 = α subunit, exons 12–22 = β subunit) [56]. The INSR gene encodes a 190 kDa proreceptor that undergoes a number of processing steps. N-linked glycosylation and amide-linked acylation occur as cotranslational events. Next, the single-chain polypeptide precursor is post-translationally cleaved in the endoplasmic reticulum into the α and β subunits. A simplified diagram showing the INSR structure is presented in Figure 3. The alternative splicing of exon 11 encodes a 12-amino acid sequence at the C-terminus of the α subunit in the IR gene during transcription, resulting in the formation of the isoforms IR-A and IR-B [54,57]. IR-B is a mature isoform because it includes the 12-amino acid sequence, whereas the fetal isoform IR-A does not [58,59]. Although both isoforms have a similar affinity for insulin, IR-A exhibits a higher affinity for IGFs, especially IGF-II, as well as greater fluctuations in its expression than IR-B [60,61]. These isoforms present different functional features and are co-expressed in most cell types, where they can form homodimers (i.e., INSR-A/INSR-A and INSR-B/INSR-B) and heterodimers (i.e., INSR-A/INSR-B) depending on the sorting of the two variants within lipid rafts (lipid microdomains) in the external leaflet of the plasma membrane. However, IR-B possesses more complex and important metabolic functions associated with metabolic and differentiating signals and is the dominant isoform in physiologic conditions [59]. Conversely, IR-A mainly favors cell growth, proliferation, and survival [62].

Subunits α and β then undergo an ester-linked acylation step, N-linked complex glycosylation, and O-linked glycosylation (β subunit), followed by dimerization. The mature insulin receptor is inserted into the plasma membrane and can be further regulated on the cell surface by insulin binding and receptor-mediated endocytosis. Thus, the INSR concentration on the cell surface or the INSR density in the tissue is a function of the internalization rate and the receptor recycling rate [57,63].

**Figure 3 ijms-24-09818-f003:**
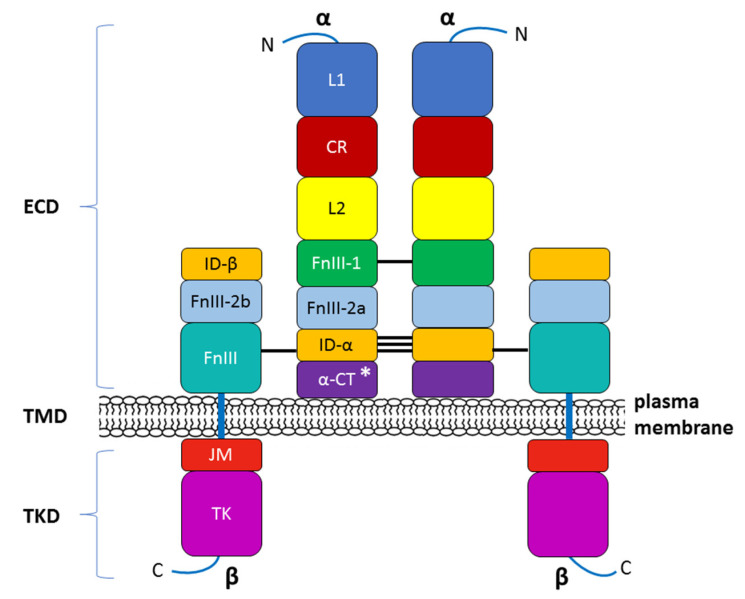
Modular structure of the insulin receptor (INSR). The INSR is a preformed, covalently linked tetramer with two extracellular α subunits and two cell membrane-penetrating, tyrosine kinase-containing β subunits. The transmembrane localization of the receptor corresponds to the presence of the ectodomain (ECD) and the transmembrane (TMD) and tyrosine kinase (TKD) domains. The binding of insulin or IGF-1 to the dimeric ECD triggers auto-phosphorylation of the TKD and subsequent activation of downstream signaling molecules. On the left half of the INSR, spans of recognized protein modules whose boundaries typically correspond to the boundaries of the 22 exon-encoded sequences detected within the α and β subunits (not shown in this figure). N—NH2-terminus of the polypeptide chain; L1, L2—two large, leucine-rich repeat domains; CR—cysteine-rich region (domain); FnIII-1, FnIII-2a, FnIII-2b, FnIII-3—fibronectin type III domains; ID-α, ID-β—α or β subunit insert domain in FnIII-2 (the splitting into FnIII2a and FnIII2b), respectively; α-CT—C-terminus of the α subunit; JM—juxtamembrane domain; TK—tyrosine kinase domain; C—C-terminal tail. The two α subunits are linked by a disulfide bond between the two Cys 524 in FnIII-1. One to three of the triplet Cys at 682, 683, and 685 in the ID-α within FnIII-2 are also involved in α–α disulfide bridges. There is a single disulfide bridge between the α and β subunits between Cys 647 in the ID-α and Cys 872 in FnIII-3 (nomenclature of the B isoform) [64]. * The alternative splicing of exon 11 encodes a 12-amino acid sequence at the C-terminus of the α subunit in the INSR gene during transcription, resulting in the formation of the isoforms INSR-A and INSR-B.

### 1.3. INSR Signaling

Following the binding of insulin or IGF-1 molecules at the binding site in the ectodomain of the INSR within the α chains, INSR activation (as well as IGFR1 activation) initiates a cascade of phosphorylation events that leads to the activation of enzymes that control many aspects of metabolism and growth [64]. The β subunit in the INSR increases tyrosine kinase activity, resulting in autophosphorylation that promotes the aggregation of heterodimers and stabilizes the receptor tyrosine kinase-activated state [55,65,66]. Next, a complex intracellular signaling network corresponding to the canonical pathway includes the phosphorylation of the docking proteins performing an adaptor function, that is, the insulin receptor substrates (IRSs). The canonical pathway is dichotomized into two branches that correspond to the phosphoinositide 3-kinase/protein kinase B/mammalian target of rapamycin (PI3K/AKT/mTOR) and mitogen-activated protein kinase/extracellular signal-regulated kinase (MAPK/ERK) pathways [64,67]. These signaling cascades containing phosphorylation products acting as insulin and IGF-1 second messengers are central to mediating the actions of both the INSR and IGFR1 in all tissues, including the liver. The specificity of signaling between insulin and IGF via the INSR and IGFR1 is probably caused by features intrinsic to the ligands or receptors themselves. It is generally accepted that insulin exerts largely “metabolic” effects, whereas IGFs are responsible mainly for “growth-promoting” effects [67]. As a consequence of INSR activation, glucose transporters translocate from sequestered sites to become exposed on the cell surface [67,68]. The insulin–INSR complex is then internalized into the endosomal apparatus in the cell [69,70]. The main signaling proteins involved in insulin signal transduction within the canonical pathway are presented in Figure 4.

The other known INSR ligands that are not classified as IRSs, such as Grb2-associated binder (Gab) family proteins, β-arrestins, cytohesins, receptor for activated C kinase 1 (RACK1), downstream of kinases, adapter proteins SH2B, and Crk, as well as the other pathways (nonreceptor tyrosine kinases, phosphoinositide kinases, and reactive oxygen species), are much less well understood and will not be discussed in detail in this review, except in relation to IRSs [67].

#### IRS Proteins

IRSs are a family of six cytoplasmatic proteins (IRS1–6) with no kinase or other enzymatic activity that act as adaptor proteins and are phosphorylated by an activated INSR [71]. IRSs have differential tissue distribution and function [72]. IRS1 and IRS2 are ubiquitously expressed and widely distributed in most cell types and are the primary mediators of INSR-related regulation of glucose metabolism and mitogenesis in most cell types [73,74]. IRS3 was identified only in rodent brains and adipocytes, whereas none of the molecular approaches provided evidence for a functional IRS-3 gene in human tissue [75]. IRS4 is characteristic of fetal development, and its expression in humans is restricted mainly to the brain, kidney, thymus, and liver [76,77]. IRS-5 and IRS-6 are less related IRS family members (also known as docking proteins 4 and 5 (DOK4 and DOK5, respectively), which have limited expression and far fewer known functions. Both proteins share homology in their N-termini, and although truncated at their C-termini, preserved functionalities in INSR signaling [78,79,80]. IRS protein sequence alignment analysis revealed high levels of homology in the N-terminal regions, where two conserved domains required for receptor recruitment are located: the pleckstrin homology (PH, also known as IH1) domain and the phosphotyrosine binding (PTB, also known as IH2) domain. The PH domain participates in protein-protein interactions and facilitates recruitment by receptors and phospholipid proteins located in the plasmatic membrane, whereas PTB contains the tyrosine residues that bind to the phosphorylated NPXY motif (Tyr-960) in the activated insulin receptor, providing a specific mechanism for the interaction between the receptor and IRS-1 [81,82,83,84]. The activation of IRS proteins takes place via the phosphorylation of approximately twenty tyrosine residues within the C-terminal region. Phosphorylated IRSs can bind to tyrosine kinase signaling protein modules of approximately 100 amino acids, the so-called Src homology (SH2) domains. The SH2 domains recognize residues C-terminal to the phosphotyrosine in their cognate peptide target ligands, including PI3K (precisely, the PI3K p85 subunit), growth factor receptor-bound protein 2 (GRB2), Src homology region 2-containing protein tyrosine phosphatase 2 (SHP2), and other less extensively studied effector proteins such as Fyn, c-Crk, CrkII, and Nck [72,81]. The binding of an SH2 domain to its cognate tyrosine-phosphorylated target links receptor activation to multiple downstream signaling pathways, both to the nucleus to regulate gene expression and throughout the cytoplasm of the cell [85,86,87]. The crucial function of IRSs is therefore the transduction of signals from the extracellular to the intracellular environment via transmembrane receptors. Tyrosine phosphorylation of IRS proteins initiates the canonical pathway, i.e., triggers the activation of both PI3K/AKT/mTOR and MAPK/ERK signaling through IRS binding to the PI3K p85 subunit and GRB2 (as well as Nck, Crk, or the Fyn kinase), respectively. Notably, the release of insulin from pancreatic cells fully relies on this pathway, and IRS-2 is fundamental for insulin signaling in hepatocytes and β-cells [88,89,90]. As the major molecules that mediate the response to insulin and IGF-1 stimulation, as well as responding to other hormonal stimuli and steroids, cytokines, and integrins, IRSs regulate many processes, including metabolism, growth, normal and cancer cell proliferation, and survival [71,91]. Following activation by cytokine and hormone receptors (e.g., IL-4, leptin, and angiotensin), IRS proteins are able to use Janus kinase 2 (JAK2) and the IRS/JAK2 interaction to activate JAK2/STAT3 with further induction of PI3K/AKT/mTOR and MAPK/ERK signaling [92,93].

It was documented for IRS1 that, considered typical cytosolic proteins, IRSs may contain native nuclear localization signals (NLSs) and can be translocated to the nucleus, probably via association with other NLS-equipped proteins [94,95]. The presence of nuclear IRS-1 was confirmed in cells expressing human JC virus T-antigen, SV40 T-antigen, integrins, estrogen receptor α (ERα), and estrogen receptor β (ERβ) [94,95,96,97]. The role of nuclear IRSs remains unclear. It cannot be ruled out that this IRS location relates exclusively to IGFR1, not INSR [98,99].

The stability of IRS proteins is regulated bidirectionally primarily by ubiquitination through several E3 ubiquitin ligases that increase IRS stability and deubiquitination by deubiquitinating enzymes (DUBs, also known as deubiquitinating peptidases) that precede IRS degradation by the proteasome. Interestingly, prolonged insulin/IGF stimulation induces both ubiquitination and degradation (deubiquitination) of IRSs. This well-documented phenomenon creates one of the major negative-feedback loop mechanisms for insulin/IGF signaling [100,101,102].

Serine phosphorylation is another mechanism for IRS protein downregulation based on the negative feedback mechanism in the insulin/IGF signaling network [64]. Given that IRS1 has over 70 potential phosphorylation sites activated by insulin, IRS kinases such as ERK, ribosomal S6 kinases (S6K1 and S6K2), and c-Jun-N-terminal kinases (JNKs) may act effectively [103]. Moreover, serine phosphorylation of IRS1 via JNKs and inhibitor of NF-κB (IκB) kinases (IKKs) is involved in tumor necrosis factor-alpha (TNF-α)/TNF-α receptor 1 (TNFR1) signaling, which is not limited to inflammation and apoptosis [86,104]. It has been shown in human, animal, and in vitro studies that such immunometabolic messenger crosstalk between TNFR1- and INSR-related pathways may be responsible for the induction of IR, especially in obesity and T2D [104,105,106,107]. In addition to diabetes, deregulation or metabolic reprogramming of IRSs has been implicated in cancer [71,72,77,108].

## 2. Insulin Resistance (IR)

IR, also known as impaired insulin sensitivity, is defined as an impaired biological response to insulin stimulation of target tissues, including primarily the liver, muscle, and adipose tissue (AT) [109,110]. This means that in practice, a known quantity of exogenous or endogenous insulin works much less effectively than in healthy people at increasing glucose uptake and utilization to preserve glucose homeostasis [111]. In a clinical setting, it has been arbitrarily assumed that patients requiring greater than 1 unit/kg/24 h of exogenous insulin to maintain glycemic control and prevent ketosis are insulin resistant [112,113]. Due to IR, the compensatory mechanism is activated, in which pancreatic β-cells increase insulin production, causing hyperinsulinemia. A vicious cycle appears because increased anabolic activity caused by the overproduction of endogenous insulin results in weight gain, which, in turn, exacerbates IR [114]. After approximately 10 to 15 years of IR, decompensation leads to relative insulin deficiency with consequent overt diabetes, classified as T2DM. Apart from the dominant result of IR, which is T2DM, the metabolic consequences of IR include a broad spectrum of diseases and pathologic conditions, such as hypertension, dyslipidemia, visceral adiposity, nonalcoholic fatty liver disease (NAFLD), ovarian dysfunction and hyperandrogenism in women, acromegaloid features, hyperuricemia, elevated inflammatory markers, endothelial dysfunction, a prothrombic state, cancer, and neurodegenerative disorders, including AD [115,116,117,118]. Consequently, IR is a permanent component of metabolic syndrome, a cluster of conditions that occur together, increasing the risk of heart disease, stroke, and T2DM [119]. In addition, an early and common manifestation of severe IR is skin changes known as acanthosis nigricans that usually develop in skin folds, such as the back of the neck, axilla, and groin, and involve darkening and thickening of the skin (velvety overgrowth of the epidermis) [116]. The pathogenesis of acanthosis nigricans is closely related to hyperinsulinemia. High circulating insulin levels cross-react with IGFR1 on keratinocytes and dermal fibroblasts and may displace IGF-1 from IGF binding protein. Increased circulating IGF-1 levels result in the excessive proliferation of keratinocytes and dermal fibroblasts [120,121].

When referring to the causes of IR, abnormalities in INSR function related to defects in receptor structure (i.e., alternative splicing and gene polymorphisms), expression, binding affinity, and/or signaling capacity (e.g., the level of tyrosine phosphorylation) should be taken into account. Moreover, the molecular mechanisms for IR also include insulin receptor antibodies, gene polymorphisms, and negative regulation of IRSs, as well as negative regulation of PI3K/AKT/mTOR and MAPK/ERK signaling. All of the above mechanisms are extensively reviewed in the cited references [122,123,124,125,126,127,128,129,130].

Hyperglycemia itself also accounts for the development of IR because a coexisting proinflammatory environment leads to the generation of reactive oxygen species (ROS), especially in response to TNF-α, which weakens insulin-induced tyrosine autophosphorylation of the INSR [131,132]. Moreover, activation of tumor necrosis factor receptor 1 (TNFR1)-mediated proinflammatory pathways impair insulin signaling at the level of the IRS proteins. In this regard, the Ser307 residue in IRS-1 has been identified as a site for the inhibitory effects of TNF-alpha in myotubes, with p38 mitogen-activated protein kinase and inhibitor kB kinase being involved in the phosphorylation of this residue. Unlike in the brown adipocytes, where the mechanism for Ser phosphorylation of IRS-2 mediated by TNF-α activation of mitogen-activated protein kinase is predominant. Protein-Tyr phosphatase (PTP)1B acts as a physiological, negative regulator of insulin signaling by dephosphorylating the phosphotyrosine residues of the INSR and IRS-1, and PTP1B expression is increased in muscle and white adipose tissue of obese and diabetic humans and rodents [133]. Importantly, for jamming INSR signaling, TNF-α is overexpressed in the adipose tissues of obese animals and humans [104]. As highly oxidant molecules, ROS can oxidize various intracellular components, including membrane phospholipids, proteins, and DNA. Usually, these reactions lead to cellular damage or a significant reduction in oxidized biomolecules. In IR, increased ROS production and/or decreased ROS degradation were demonstrated, leading to an oxidative stress condition with subsequent activation of signaling pathways related to stress [132]. Considering the above information, one can speculate that glucose intolerance and IR may serve as protective mechanisms that prevent glucose toxicity.

The main underlying causes of IR are summarized in Figure 5.

Despite a significant increase in knowledge about the various causes of IR, a precise explanation of the molecular mechanisms resulting in the development of this disorder has still not been elucidated [141]. Current research on IR includes, among others, intracellular redox balance, oxidative and nitrative stress, mitochondrial dysfunction, increased plasma-free fatty acid (FFA) levels, especially long-chain saturated fatty acids (SFAs), and the accumulation of intracellular lipid derivatives (diacylglycerol and ceramides) [142,143,144,145]. Mutations and the expression (modulation of transcription) of genes involved in lipid and glucose metabolism as well as insulin signaling, inflammation, redox balance, and mitochondrial function are being studied [134]. As gene expression is regulated at various levels and not only in response to DNA modifications, the role of epigenetic mechanisms in IR might be worth noting, including histone modifications (e.g., acetylation and deacetylation), altered DNA methylation of insulin or INSR genes, and noncoding RNAs (ncRNAs), i.e., microRNAs (miRNAs) and long noncoding RNAs (lncRNAs) [146,147]. For example, the long noncoding RNAs (lncRNAs) MEG3 and H19 are involved in obesity and IR in women. The results showed lower mRNA levels of H19 in subcutaneous adipose tissues (SATs) of obese women compared to normal-weight women and a significantly higher expression of MEG3 in the SAT of the obese group vs. lean controls [148].

The relationship between IR and the accumulation of lipids in tissues suggests that these lipids are both markers and mediators of metabolic dysfunction, especially in skeletal muscle, which is the main site of insulin-stimulated glucose disposal [149,150]. Incorrect adjustment to the glucose-fatty acid cycle (also called the Randle cycle) may therefore be the cause of IR. Normally, this biochemical phenomenon is based on the competition between glucose and fatty acids for their oxidation and uptake in muscle and adipose tissue. Briefly, the oxidation of fatty acids yields acetyl-CoA, from which citrate is generated by the action of citrate synthase. Next, increased values of the acetyl-CoA/CoA and NADH/NAD+ ratios stimulate pyruvate dehydrogenase kinase (PDK), which phosphorylates and inactivates pyruvate dehydrogenase (PDH). ATP and citrate inhibit phosphofructokinase with the accumulation of substrates from previous stages, including glucose-6-phosphate (G-6-P), which has an inhibitory effect on hexokinase, ultimately repressing glycolysis [149]. As a result, a significant reduction in the uptake and utilization of glucose occurs when fatty acid oxidation is intensified. This glucose-fatty acid crosstalk is also observed in AT [151]. It was recently proposed that the starting point for malfunction in the glucose–fatty acid cycle and related IR may be an increase in global protein acetylation from acetyl-CoA. It was demonstrated in cultured adult rat cardiomyocytes that leucine, ketone bodies, palmitate, and oleate promote a rapid increase in protein acetylation, especially in the abundance of the fatty acids that generate high levels of acetyl-CoA. Thus, enhanced protein acetylation initiates fatty acid-mediated inhibition of cardiac glucose transport, resulting in impairment in the translocation of vesicles containing the glucose transporter GLUT4 to the plasma membrane [152].

## 3. Inflammatory Response and IR (See Also the Summary in Figure 6 at the End of This Section)

The results of numerous experiments have demonstrated that in normal conditions, insulin exerts glucose homeostatic and anti-inflammatory effects. Pretreatment with insulin inhibits the activation of NF-κB and the expression of proinflammatory cytokines to alleviate the inflammatory response in vitro and in vivo [2,153,154]. The molecular mechanism for this immunomodulation provided by insulin is based on downregulation of the nucleotide-binding oligomerization domain (NOD), leucine-rich repeat (LRR)-containing protein (NLR) family member 3 (NLRP3) inflammasome, a critical component of the innate immune system that mediates caspase-1 activation and the secretion of the proinflammatory cytokines IL-1β/IL-18 in response to microbial infection and cellular damage [155]. In the case of IR, the anti-inflammatory effect of insulin is significantly hampered and insufficient to block the low-grade inflammation that typically develops, especially in obese individuals [156]. The mechanism for the vicious cycle is that IR due to insulin signaling inhibition results in a series of immune responses caused by abnormal glucose tolerance that exacerbate the inflammatory state, which in turn leads to an increase in both IR and blood glucose levels.

Abnormally high deposition of visceral AT or increased AT surrounding the intra-abdominal organs is known as visceral obesity. It has been distinctly linked to several pathological conditions, including impaired glucose and lipid metabolism and IR, and is an independent component of metabolic syndrome [157]. Such an excess of fat tissue alters the secretion of FFAs, adipokines, growth factors, and proinflammatory cytokines. Moreover, obesity increases macrophage infiltration within the AT, which can increase the level of proinflammatory cytokines. Altered glucose and lipid metabolism are also clearly detectable in hepatic, pancreatic, and muscular tissues as the modifications to the inflammatory response spread out. Thus, the contribution of the inflammatory process to IR is not only limited to AT but is also generalized (systemic) [158]. Importantly, among the proinflammatory factors secreted in the AT and by macrophages are those that can directly contribute to IR, including TNF-α, IL-1β, IL-6, IL-18, and angiotensin II (Ang II). For example, it was suggested that Ang II may lead to IR through protein kinase C (PKC) activation in adipocytes [159]. In the case of TNF-α, IL-1β, IL-6, and IL-18, the impaired biologic response to insulin stimulation of target tissues may be caused by multiple mechanisms, such as degradation of IRS1, serine/threonine protein kinase (STK) activation, impaired GLUT4 translocation from intracellular stores to the plasma membrane, decreased expression of peroxisome proliferator-activated receptor gamma (PPAR-γ), or increased expression of suppressor of cytokine signaling protein 3 (SOCS-3) [160,161,162,163,164,165,166].

The innate and adaptive immune systems are involved in the pathomechanism for obesity-induced inflammation and systemic proinflammatory polarization of immune cells, resulting in augmented secretion of proinflammatory cytokines [167].

Considering the involvement of CD4+ T lymphocytes depending on the stimuli/environments, the inflammatory response can be broadly divided into two types: type 1, related to T helper 1 (Th1) lymphocytes, and type 2, related to Th2 cells. In simple terms, Th1 cells promote and coordinate the cell-mediated host inflammatory response to intracellular pathogens by inducing the activation of macrophages, NK cells, B cells, and CD8 T cells, whereas Th2 cells mediate the activation and maintenance of the humoral, or antibody-mediated, immune response against extracellular pathogens. Th1 cells primarily synthesize interferon-gamma (IFN-γ), TNF-α, and interleukin-2 (IL-2), while Th2 cells synthesize other cytokines, such as IL-4, IL-5, IL-6, IL-9, IL-13, and IL-17E (IL-25). Suited to the immune challenge, well-balanced Th1 and Th2 responses are required in humans because it was demonstrated that the clear dominance of Th2 responses will counteract Th1-mediated microbicidal action, and vice versa [168,169].

Given that there is a linking mechanism between obesity, inflammation, and IR, the predominance of type 1 inflammatory response in AT may be involved in the etiopathogenesis of altered insulin signaling. In contrast to AT type 1 inflammation in obesity, type 2 inflammatory conditions are a part of AT immune homeostasis regulation in lean healthy individuals [119,170]. It has been suggested that the initiation of generalized low-grade inflammation in obesity may be triggered by the production of chemokines by adipose cells, the release of damage-associated molecular patterns (DAMPs) from necrotic AT, augmented rates of lipolysis with a subsequent increase in the flux of nonesterified fatty acids, and hypoxia with activation of hypoxia-induced factor (HIF)-1, which controls the expression of proinflammatory proteins [171].

### 3.1. Innate Immune System and Pattern Recognition Receptors (PRRs)

The innate immune system, or nonspecific immune system, is the body’s first line of defense against germs entering the body. This system includes physical and anatomical barriers as well as effector cells, humoral mediators, antimicrobial peptides, and a limited number of diverse receptors that are encoded by intact genes inherited through the germline, that is, pattern recognition receptors (PRRs) [172]. PRRs are proteins expressed mainly on the effector cells of the innate immune system, such as macrophages, monocytes, granulocytes (especially the most abundant neutrophils), dendritic cells, and epithelial cells, which are capable of identifying two classes of molecules: pathogen-associated molecular patterns (PAMPs), which are frequently found in microbial pathogens, and DAMPs, which are unique molecules displayed on or released from stressed, injured, infected, or transformed cells. There are several subgroups of PRRs. They are classified according to their ligand specificity, function, localization, and/or evolutionary relationships. Based on their localization, PRRs may be divided into membrane-bound PRRs, including Toll-like receptors (TLRs) and C-type lecithin receptors (CLRs), and cytoplasmic PRRs, including nucleotide oligomerization domain (NOD)-like receptors (NLRs) and retinoic acid-inducible gene I (RIG-I)-like receptors (RLRs) [173]. Among all the PRRs identified, TLRs are the most ancient class, with the most extensive spectrum of pathogen recognition [174,175].

Ligation of PRRs on immune cells leads to the transmission of a threat signal to the cells and initiates a cascade of responses that direct host defense responses. Typically, the response is based on cytokines or antimicrobial compounds produced by leukocytes after stimulation by PRRs. The second important consequence of PRR detection is the induction of competency in selected cells to present antigens to T cells. Such maturation of antigen-presenting cells (APCs) relies on the stabilization of major histocompatibility complex (MHC) molecules on their surface. Thus, PRRs are also important for the initiation of adaptive immunity because the presentation of antigens by APCs is necessary to initiate adaptive immune responses [176].

As obesity-related insulin resistance develops, adipocyte cell death and innate immune activation provide antigenic stimuli similar to those in microbial or other infections. DAMPs, consisting of endogenous intracellular molecules released by necrotic cells and extracellular matrix (ECM) molecules that are upregulated upon injury, provide a clear signal of vital danger. Then, TLRs are activated, resulting in increased inflammatory gene expression to mediate tissue repair processes [177].

Transmembrane proteins known as TLRs belong to the family of Type I integral membrane glycoproteins, and in humans, they comprise a family of ten members (TLR1–TLR10) responsible for detecting various PAMPs and DAMPs. TLRs are expressed on all innate immune cells (innate leukocytes), such as natural killer (NK), mast cells (MCs), eosinophils, and basophils, as well as on phagocytic cells, including macrophages, neutrophils, and dendritic cells (DCs). Moreover, TLRs can also be detected on adaptive immune cells, including T and B cells, and a large majority of nonhematopoietic cells, such as mesenchymal stromal cells, fibroblasts, and endothelial cells [178]. Thus, molecules released following TLR activation signal to other cells in the immune system, making TLRs key elements of innate immunity and adaptive immunity. The dynamic nature of TLR expression is that modulation of the expression pattern may rapidly change in response to environmental stresses, pathogens, or a variety of released cytokines. Furthermore, TLRs may be expressed on the cell surface (extracellularly), as in the case of TLRs 1, 2, 4, 5, and 6, or may almost exclusively be in intracellular compartments such as endosomes and their ligands, primarily nucleic acids. Intracellular localization was confirmed for TLRs 3, 7, 8, and 9.

Although each of the TLRs recognizes different molecular patterns, the structure of all TLRs is generally quite conserved. All TLRs contain extracellular domains containing variable numbers of leucine-rich-repeat (LRR) motifs and a cytoplasmic Toll/interleukin 1 (IL-1) receptor (TIR) homology domain. The LRR domain, responsible for ligand binding, is connected via a transmembrane domain with a TIR domain that is required for signal transduction. The transmembrane and membrane-proximal regions are important for the cellular compartmentalization of these receptors [175,179].

#### 3.1.1. TLR Activation

When bound to a specific ligand (PAMP or DAMP), TLRs form homodimers (e.g., TLR4) or heterodimers (e.g., TLR1, TLR2, and TLR6) and initiate signaling pathways that are very similar to the IL-1 receptor-mediated pathway for NF-κB activation [178,180]. TLR signaling is initiated when the cytoplasmic TIR domain interacts with adaptor molecules such as myeloid differentiation primary response gene 88 (MyD88). MyD88 is the canonical adaptor for inflammatory signaling pathways downstream of members of the Toll-like receptor (TLR) and interleukin-1 (IL-1) receptor families. MyD88 attracts the IL-1 receptor-associated kinase 4 (IRAK4) molecule to the TLR family members or IL-1 receptor, which induces the phosphorylation of IRAK1 and IRAK2 and the formation of the signaling complex. Following activation of IRAK family kinases, the signaling complex reacts with TNF receptor-associated factor 6 (TRAF6), which leads to TGFβ-activated kinase 1 (TAK1) activation and consequently the induction of inflammatory cytokines via nuclear factor κB (NF-κB), MAPKs, and activator protein 1 (AP-1) [181]. Thus, the adaptor molecule MyD88 serves as a central node in inflammatory pathways [182,183].

In addition to the MyD88-dependent pathway, TLRs may be activated through Toll/IL-1R domain-containing adaptor-inducing IFN-β (TRIF)-dependent signaling. The TRIF-dependent pathway is induced by macrophages and DCs after TLR3 and TLR4 stimulation. This pathway is required for the TLR-mediated production of type-I IFN and several other proinflammatory mediators [184].

TLR signaling in IR, especially coexisting with obesity, may play a pivotal role in the development of so-called trained immunity, a functional state of the innate immune response characterized by long-term epigenetic reprogramming of innate immune cells, which depends on prior exposure to a signal [158,185]. This reprogramming of monocytes, NK cells, and DCs is a manifestation of an adaptation in the innate immune system to the metabolic changes related to altered insulin signaling because the factors demonstrated to facilitate trained immunity include, in addition to pathogenic signals (bacteria, viruses, and fungi), initially nonpathogenic signals such as insulin, cytokines, adipokines, or hormones [186].

It was demonstrated that TLR4 contributes to the development of IR and inflammation in obesity via its activation by excess enteric lipopolysaccharide (LPS) and SFAs as well as via endogenous ligands, e.g., FFAs. The resulting activation of the proinflammatory kinases JNK, IKK, MAPKs, and p38 in insulin target tissues impairs insulin signal transduction directly through inhibitory phosphorylation of IRSs on serine residues [187]. The tissue inflammatory state during obesity may be caused by TLR-4/NF-κB pathway activation in macrophages via FFA action, which promotes the synthesis and secretion of proinflammatory cytokines, including IL-6, TNF-α, IL-1β, and IL-18 [188].

Activation of the TLR4 pathway promotes the priming of the NLRP3 inflammasome, a multiprotein complex that assembles in the cytosol after exposure to PAMPs or DAMPs [189] (see the next Section 3.1.2).

#### 3.1.2. NLR Family Pyrin Domain Containing 3 (NLRP3) Activation

The activation of NLRPs results in the formation of many distinct inflammasome complexes, each of which has a unique PRR and activation trigger [189]. The best characterized is the NLRP3 complex, which is composed of NLRP3, apoptosis-associated speck-like protein containing caspase recruitment domain (ASC, also PYCARD), pro-caspase-1, and the serine-threonine kinase NEK7 (NIMA “never in mitosis gene A”-related kinase 7). NLRP3 is a cytoplasmic PRR that acts as a dominant innate immune sensor for tissue damage. For that reason, it plays a crucial role in the development of sterile inflammation during the course of AT metabolic dysfunction caused by IR [190,191]. By sensing self-danger signals (activation by DAMPs), NLRP3 undergoes a conformational change (oligomerization) that results in unfolding and binding to the key adaptor protein ASC through homotypic pyrin–pyrin domain interactions, ultimately leading to ASC nucleation. On this scaffold, recruitment, self-cleavage, and autoactivation of the effector pro-caspase-1 generate mature caspase-1 and eventually lead to NLRP3 inflammasome formation. Next, activated caspase-1 processes pro-IL-1β and pro-IL-18 into their bioactive forms to initiate inflammation [192,193]. Thus, inflammasomes are molecular platforms that are activated upon cellular infection or stress, including IR-induced metabolic dysregulation, that trigger the maturation of proinflammatory cytokines [194]. Strong associations between dysregulated inflammasome activity and human heritable and acquired inflammatory diseases highlight the importance of this pathway in tailoring immune responses [189,193].

### 3.2. Inflammatory Response Related to B Cells and T Cells

The adaptive, or acquired, immune response to an antigen is more specific but takes days or even weeks to become established—much longer than the innate immune response. Moreover, without information from the innate immune system, the adaptive response cannot be mobilized [195]. Two main types of adaptive immune responses, humoral and cell-mediated responses, are strictly linked to the function of lymphocyte subtypes called B cells and T cells, respectively. Activated B cells proliferate and differentiate into antibody-secreting effector cells, known as plasma cells. Apart from generating antibodies and an antibody-mediated memory response against pathogens, B cells secrete cytokines and are also capable of generating cell-mediated immunity, serving as professional APCs to activate antigen-specific CD4+ and CD8+ T cells. CD4+ T cells, which mature in the thymus gland, are MHC-II restricted and preprogrammed for helper functions, whereas MHC I-restricted CD8+ T cells are preprogrammed for cytotoxic functions and are capable of detecting immunogenic peptide–MHC class I (pMHCI) complexes presented on nucleated cells [196]. Specific CD4+ T cells are activated and undergo clonal expansion after the recognition of antigenic peptide/MHC-II on antigen-presenting cells via a T-cell receptor [197,198]. Following activation by recognized antigenic structures, CD4+ T cells activate B cells and differentiate into distinct subpopulations that produce different cytokines. Similarly, CD8+ T cells mediate their effector functions with the production of cytokines such as IFN-γ and tumor necrosis factor (TNF)-α and/or by cytolytic mechanisms [199].

IR significantly affects the adaptive immune system, although the root cause is unclear. It cannot be excluded that in this interdependence, the IR initiating factor comes from the adaptive immune system and is associated with a low-grade chronic inflammatory state within excessive AT. Although macrophages are the most abundant immune cell type in the AT of both mice and humans, and the M1 phenotype might especially be a major driver of AT inflammation, T cells and B cells also play important roles in modulating AT inflammation [200,201,202,203].

Interestingly, as IR is primarily associated with obesity, prediabetes, and T2D, people with type 1 diabetes (T1D) can also become insulin resistant. IR is not a cause of T1D, but people with type 1 insulin resistance will need higher insulin doses to keep their blood glucose under control than those who are more sensitive to insulin. The autoimmune process mediated via T cells is a main part of the etiopathogenesis in T1D, but B cells also clearly participate in pancreatic islets of Langerhans destruction, as autoantibodies recognizing insulin-secreting β cell antigens commonly appear in the circulation before the onset of the disease [204]. The development of IR in T1DM may be triggered by many risk factors, and even if it is distinctively different from the IR in T2DM, obesity and the inflammatory background are consistently present in both types of diabetes [205].

#### 3.2.1. B Cells

Immunological B cells are generally divided into two major subsets. B2 cells generate specific antibodies against foreign antigens in secondary lymphoid organs. B1 cells, found predominantly in the peritoneal and pleural cavities, instead produce “natural” antibodies as part of the innate immune system. B cells that reside in nonlymphoid organs, including AT, function to maintain homeostasis at a steady state [206,207].

It was demonstrated that B cells can modulate AT function in obesity. Moreover, an increase in the proportion of B cells is also a feature of adipose tissue in obesity. B cells are one of the first immune cells to accumulate in AT in response to a high-fat diet. A three-fold increase in B cell number in epididymal visceral AT depots was reported in C57BL6/J mice three weeks after a high-fat diet [208]. Moreover, diet-induced obese (DIO) mice lacking B cells are protected from IR despite weight gain due to the accumulation of AT. Consequently, treatment with a B cell-depleting CD20 antibody attenuates IR, whereas transfer of IgG from DIO mice rapidly induces IR and glucose intolerance [209]. In visceral AT, B cells are involved in the activation of proinflammatory macrophages and T cells as well as in the production of pathogenic IgG antibodies. The profile of IgG autoantibodies isolated from obese humans with IR is specific and differs from that of nonobese diabetic persons or obese nondiabetic persons [209]. It was suggested that a change in the proportion of B1 to B2 cells in obese AT resulting from the increase in B2 cells may induce IR. The recruitment/chemotaxis of B2 cells to AT and direct activation or stimulation of the B2 cell proinflammatory phenotype are processes mediated by signaling through chemokine leukotriene B4 (LTB4) and its G protein-coupled receptor, LTB4R1 (also known as BLT1). LTB4 is an endogenous lipid mediator of inflammation derived from arachidonic acid by the sequential action of cytosolic 5-lipoxygenase, 5-lipoxygenase-activating protein, and leukotriene A4 hydrolase. LTB4, via LTBR1, induces the recruitment and activation of neutrophils, monocytes, and eosinophils. It also stimulates the production of a number of proinflammatory cytokines and mediators, indicating an ability to augment and prolong tissue inflammation [210,211,212].

Loss of LTB4R1 prevents obesity-induced B2 cell recruitment into visceral adipose tissue, mitigating the contribution of B2 cells to the pathogenesis of obesity-induced adipose tissue inflammation and IR. Thus, the LTB4/LTB4R1 axis facilitates the metabolic effects of B2 cells toward glucose intolerance and IR in obese AT. Moreover, an increased number of B2 cells in obese AT exacerbates insulin resistance through Th1 lymphocyte- and M1 macrophage-mediated mechanisms. In addition, previous studies have demonstrated the effects of the LTB4/LTB4R1 axis on the recruitment and activation of macrophages in the context of obesity [213,214,215,216].

Thus, the recruitment of B lymphocytes in obese visceral AT precedes the infiltration of inflammatory CD4+ and CD8+ T cells and macrophages, leading to Th1 polarization and the production of IgG autoantibodies by B2 cells, which are major factors contributing to the development of IR [208,209,216,217].

It is important to note that, unlike B2 cells, B1a and B1b cell subtypes are able to improve insulin sensitivity in diet-induced obesity with the secretion of IgM autoantibodies. In addition, distinct B cell populations, designated regulatory B (Breg) cells, produce the anti-inflammatory cytokine CXCL12 (C-X-C Motif Chemokine Ligand 12), which maintains the balance of the immune response and restrains immune responses associated with autoimmune diseases. Adipose environmental factors, including CXCL12 and free fatty acids, support Breg cell function, and the Breg cell fraction and function were reduced in adipose tissue from obese mice and humans. Therefore, B1a, B1b, and Breg are promising therapeutic targets in obesity to overcome IR [218,219,220].

Regardless of high-fat diet consumption, the aging of the body has been linked to AT B cell mediation via visceral AT accumulation of follicular B2 cells and “age-associated B cells” (ABCs) that can produce proinflammatory IgG and cytokines. ABCs are a heterogeneous B cell subset (CD19+, CD21−, CD11c+, T-bet+) that is expanded in the elderly but also accumulates prematurely in patients with autoimmune disorders and/or infectious diseases [221,222]. Moreover, the expression of a B cell-specific coactivator of octamer-binding transcription, Oct coactivator B (OcaB), is increased in visceral AT with age in both humans and mice. OcaB is important for the transcription of variable parts of the kappa light chain of immunoglobulins. Global deletion of OcaB in mice resulted in a reduction in B2 cell numbers in AT and reduced proinflammatory IgG2c antibody levels with subsequent improvement in insulin sensitivity [223].

It was recently demonstrated that phenotypic and functional features of B cells from two different human subcutaneous adipose depots may differ significantly [224]. Analyses of the B cells from the breast and abdominal subcutaneous AT revealed that in obese women, B cells from the abdominal AT are “more inflammatory” than those from the breast. This inflammation manifested itself as higher frequencies of inflammatory B cell subsets, higher expression of RNA for inflammatory markers associated with senescence and higher secretion of autoimmune antibodies in abdominal AT than in breast AT. Moreover, a higher number of autoimmune B cells with the membrane phenotype CD21lowCD95+ B cells expressing the transcription factor T-bet (Tbx21) was found in the abdominal AT. It is worth noting that T-bet directs T cell homing to proinflammatory sites by regulating chemokine receptor CXCR3 expression [225,226]. This highlights the pivotal role of abdominal AT in the development of chronic inflammation and IR.

#### 3.2.2. T Cells

Chronic inflammation characterized by T cell and macrophage infiltration of visceral AT is a hallmark of obesity-associated IR and glucose intolerance. The cell-mediated immune response is especially present in obesity when hypertrophic adipocytes produce proinflammatory cytokines, such as IL-6 and TNF-α, which leads to increased vascular permeability and the recruitment of cytotoxic T cells, including T cell phenotypes associated with IR [227,228,229]. Indeed, insulin-sensitive (IS) patients with metabolically healthy obesity (MHO) show significant immunological differences in T cell phenotypes compared to obese patients with IR (metabolically unhealthy). For example, the frequencies of naïve (CD45RA+ CCR7+ CD27+ CD28+) CD4+ and CD8+ T cell subsets were positively associated with the ISIOGTT (insulin sensitivity index (ISI) obtained from the oral glucose tolerance test (OGTT)) and inversely correlated with the HOMA-IR (homeostasis model assessment of insulin resistance) in older participants, whereas the percentage of central memory (CD45RA− CD27+ CD28+) CD4+ T cells was negatively associated with the ISIOGTT and positively associated with the HOMA-IR. In addition, the percentage of effector memory (CD45RA− CD27− CD28−) CD8+ T cells correlated positively with the HOMA-IR [230]. Considering the above results, the percentage of naïve CD4+ and CD8+ T cells may be treated as a predictor for impaired insulin sensitivity. These findings support the hypothesis that parameters of systemic inflammation related to T cell subtypes can differentiate IS from IR obese individuals who are at higher risk of cardiometabolic diseases and should be preferentially subjected to obesity treatment.

Significant changes in adaptive immunity have been reported after bariatric surgery, where the most obvious effect, a loss of up to half of the total adipose tissue mass within the first year following surgery, is accompanied by a reduction in CD4+ and CD8+ T cell counts, a decrease in the Th1/Th2 ratio, an increase in B regulatory cells, and a reduction in proinflammatory cytokine secretion [228,231]. Overall, there is a shift in the T cell subtypes, from the proinflammatory to the less- or anti-inflammatory subtypes [232,233].

Alteration in the Th1/Th2 balance in obese visceral AT that is typically observed in IR may be caused by abnormal expression of the transcription factor T-bet (Tbx21) and related T cell chemotaxis governed by CXCR3 and its ligands (CXCL9 or monokine induced by IFN-γ (Mig), CXCL10, and CXCL11) [225,226]. The expression of the chemokine receptor CX3CR3 was confirmed on many different cell types, including T cells, B cells, NK cells, renal tubular epithelial cells, fibroblasts, and vascular pericytes [234,235,236,237,238]. Thus, originally described as the master regulator of Th1 development, T-bet performs regulatory functions necessary in both the adaptive and innate immune systems [226]. Quiescent T cells show weak CXCR3 expression, whereas, in the process of T cell activation, the expression of CXCR3 markedly increases. The important role of CXCR3 in supporting the proper function of activated T cells is unambiguously connected with the local inflammatory response. The addition of CXCR3 ligands to normal human T cells expressing CXCR3 led to the tyrosine phosphorylation of multiple proteins. CXCR3-mediated T cell chemotaxis involves a zeta-associated protein of 70,000 molecular weight (ZAP-70) and is regulated by signaling through the T cell receptor (TCR). This means that essential molecules in TCR signal transduction, such as ZAP-70, linker for the activation of T cell (LAT), and phospholipase-C-γ1 (PLCγ1), become phosphorylated on tyrosine 319 (ZAP-70), tyrosines 171 and 191 (LAT), and on tyrosine 783 (PLCγ1). Thus, cytoplasmic protein tyrosine kinases play a critical role in the events involved in initiating T cell responses by the TCR [239,240].

Interestingly, mice deficient in T-bet may display greater visceral AT content but paradoxically more IS than T-bet+/+ mice [241]. This finding indicates that the absence of T-bet can separate obesity from IR. However, the precise mechanism for this potentially crucial role of T-bet in physiological and pathophysiological metabolism remains unclear [226].

Peroxisome proliferator-activated receptors (PPARs) and T cells

PPARs are a group of three nuclear hormone receptor proteins (PPARα, PPARβ, and PPARγ) that promote ligand-dependent transcription of target genes that regulate energy production, glucose and lipid metabolism, cell proliferation/differentiation, and inflammation [242,243]. The human PPARα gene is located on chromosome 22, PPARβ is located on chromosome 6, and PPARγ, which encodes three isoforms, has been identified on chromosome 3 [244].

The best-known mechanism by which all PPAR subfamilies downregulate inflammation is through transrepression [245]. This activity involves indirect association (tethering) of the PPARs with target genes. There are many mechanisms by which PPARs can transrepress the inflammatory response, including competition for a limiting pool of coactivators, direct interaction with the p65 subunit of NF-κB and the c-Jun subunit of AP-1, modulation of p38 mitogen-activated protein kinase (MAPK) activity, and partitioning of corepressor B cell lymphoma 6 (BCL-6) [245,246].

PPAR natural ligands include lipid-derived metabolites such as fatty acids, acyl-CoAs, glycerol–phospholipids, and eicosanoids. PPAR polymorphisms may be responsible for the risk of being overweight or obese, whereas dominant-negative PPARγ mutations result in T2DM, hypertension, and IR [247,248]. PPARs are expressed not only in adipose tissue but also in various tissues and cell types, including pancreatic β cells and T cells, where they regulate insulin secretion and T cell differentiation, respectively [249]. PPARs have been shown to regulate T cell survival, activation, and CD4+ T helper cell differentiation into the Th1, Th2, Th17, and Treg lineages [250].

Based on human and animal studies, the anti-inflammatory role of PPARγ has been clearly demonstrated in adaptive immune cells, including T cells. For example, PPARγ activation in mouse CD4+ T cells selectively suppresses Th17 differentiation by inhibiting TGF-β/IL-6-induced expression of retinoic acid receptor-related orphan receptor γt (RORγt) [251]. Consistently, it was also demonstrated that human CD4+ T cells with PPARγ small interfering RNAs (siRNAs) increased IL-17A production [249]. However, as this effect was detected exclusively in female T cells, the results of the study revealed that human T cells exhibit a sex difference in the production of IFN-γ and IL-17A that may be driven by the expression of PPARα and PPARγ [249].

On the other hand, PPARγ is involved in the TCR–mTORC1–PPARγ and TCR–mTORC1–SREBP1 signaling axes, which are required for the activation of the fatty acid metabolic program in activated CD4+ T cells [252]. Therefore, even if PPARγ downregulates the generation of effector cells with Th17 properties, its direct binding to DNA and regulation of the induction of genes associated with fatty acid uptake and metabolism may lead to the development of memory cells related to fatty acid metabolism that may foster IR.

As demonstrated in a murine model of cardiac transplantation, T cell-specific PPARγ deficiency may lead to T cell activation and the generation of alloreactive T cells with subsequent chronic allograft rejection. This was associated with higher Thl/Th2 and Th17/Treg ratios among the infiltrating CD4+ T cells [253].

In visceral AT, PPARγ is involved in the control of the inflammatory state of adipose tissue, which translates into IS [254,255]. The control of AT inflammation by PPARγ depends on the molecular regulation mediated by PPARγ Ser273, which results in significant changes in the transcriptional profile driving the differentiation of Tregs in visceral AT [256]. By regulating lipid metabolism and calming the inflammatory response, PPARs may also influence Treg survival [257].

Overall, the improvement in/restoration of IS related to PPARγ action on metabolism may be achieved clinically using PPARγ agonists (e.g., thiazolidinediones (TZDs)) or compounds that prevent PPARγ phosphorylation (e.g., influencing ligand-independent activation function 1 (AF-1) within the N-terminal A/B domain of PPARs). A significant anti-inflammatory reprogramming of T cells is central to both therapeutic strategies [258,259,260]. The activation of PPARγ has also been suggested to regulate microRNA expression to inhibit inflammatory responses. PPARγ could upregulate microRNA (miR)-124 in vitro and in vivo to inhibit the production of proinflammatory cytokines [261].

PPARα and PPARβ are highly expressed and involved in oxidative metabolism by regulating genes that control substrate delivery and oxidative phosphorylation (OXPHOS) and the regulation of energy homeostasis. However, since they are expressed on T cells and other immune cells, PPARα and PPARβ are not as important in the regulation of the adaptive immune response in white AT as PPARγ is. Moreover, the expression levels of PPARα and PPARβ in white AT are considerably lower than the expression of PPARγ [262]. Thus, PPARα and PPARβ have an emerging critical role in immune cell differentiation and fate commitment but are not principal AT components in the context of IR [263]. Even if some of the experimental studies had linked the activation of PPARα in rodents with significant improvement in IS, the effect of a PPARα agonist in humans was much less pronounced, probably due to a lower expression of PPARα in AT relative to rodents and to possibly different mechanisms [264].
Figure 6Summary of the mechanisms by which visceral obesity is a triggering factor for both systemic inflammation and insulin resistance (IR). For details, see Section 3. Strongly enlarged adipocytes in the obese visceral adipose tissue (AT) fail to maintain longer metabolic homeostasis because the lipid overload leads to endoplasmic reticulum stress, increased expression of the inflammation regulator NF-kB, and the production of inflammation-inducing signals [171]. This chronic metabolism-induced inflammation or metaflammation in obese AT activates the resident immune cells, including macrophages, B cells, T cells, and antigen-presenting cells (APCs) [167]. A high-fat diet (HFD), increased flux of free fatty acids (FFAs) accumulating damage-associated molecular patterns (DAMPs) from necrotic AT, and hypoxia-induced factor 1 (HIF-1) trigger both innate and adaptive immune responses [171]. The resulting low-grade chronic AT inflammation manifests as significantly elevated levels of the proinflammatory adipokines and cytokines as well as through overproduction of reactive oxygen species (ROS) [6,7,8,156,157,158,159,265]. Such an inflammatory environment interferes with insulin signaling via the insulin receptor (INSR) through antibodies/autoantibodies against both insulin (INS) and INSR [125,134]; tumor necrosis factor receptor (TNFR1) through inhibition of tyrosine autophosphorylation (pY) within the INSR and/or serine/threonine phosphorylation (pS) of the insulin receptor substrates (IRSs) [122,126,133,134]; and receptors for other proinflammatory cytokines (Recs) through abnormal, similar to that observed for TNFR1 signaling, phosphorylation of the INSR and IRSs [122,126,142]. Disrupted downstream signaling via the mitogen-activated protein kinase/extracellular signal-regulated kinase (MAPK/ERK) and phosphoinositide 3-kinase/protein kinase B/mammalian target of rapamycin (PI3K/AKT/mTOR) pathways eventually lead to IR. The resulting hyperglycemia may increase the inflammatory response by itself and lead to systemic inflammation [110,111,119]. The remaining abbreviations: ANGPTL2—angiopoietin-like 2; IL-1, IL-6, IL-8, IL-12, IL-18—the respective interleukins; MCP-1—monocyte chemoattractant protein-1 (also known as CCL2); NLRP3 inflammasome—leucine-rich repeat (LRR)-containing proteins (NLR) family member 3 inflammasome; SAAs—sulfur amino acids; TGFβ—transforming growth factor β.
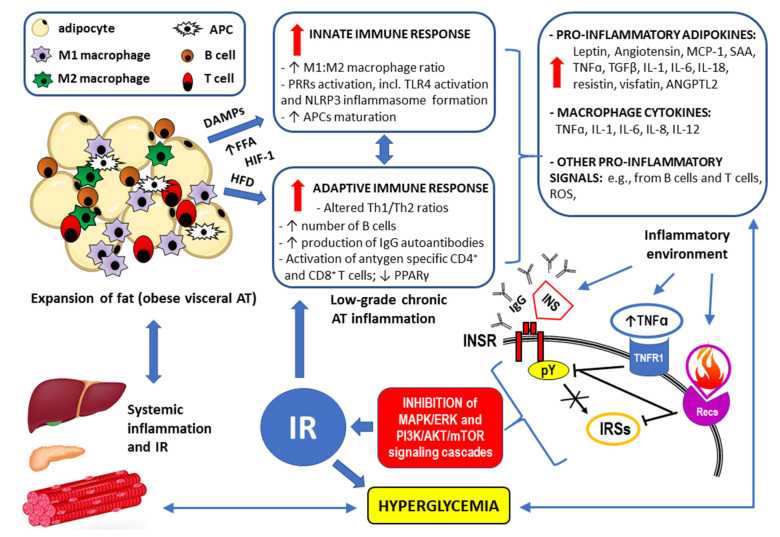


## 4. Concluding Remarks

A high prevalence of IR, especially coexisting with obesity, explains the broad clinical spectrum of dysmetabolic conditions (e.g., glucose intolerance, diabetes, and metabolic syndrome) with which the pathomechanisms for IR are associated [266]. In 1998, the WHO proposed a unifying definition for “the syndrome of abnormal metabolism” and chose to call it the metabolic syndrome rather than the IR syndrome [267]. The main reason was that it was not considered established that IR is the cause of all the components of the syndrome (e.g., hypertension, dyslipidemia, visceral obesity, or microalbuminuria) [267,268].

According to the current state of knowledge, there is no doubt that the immune response in excessive visceral AT interferes with insulin signaling. Both innate and adaptive responses take part in the development of reduced IS [167]. Given that the chronic low-grade inflammatory process in visceral AT is driven by obesity, the role of environmental factors and epigenetics should be considered [156,157]. Distinct from genetic mutation, epigenetic influences refer to modifications of gene expression (silencing or activation) without permanent changes in the genomic sequence. Epigenetic changes (e.g., DNA methylation or chromatin remodeling) provide a molecular basis for cellular memory in relation to environmental triggers [269]. Epigenetic factors are mediators of inflammation and chronic inflammatory disease. Consistently, disruption of immune cell epigenetic regulation is thus predicted to be a major contributor to unrestrained immune responses and immune tolerance disruption that lead to diseases such as autoimmunity and inflammation [270,271,272]. These epigenetic changes can be caused by smoking, exposure to chemicals, including endocrine-disrupting compounds (EDCs), in the environment, or diet and other lifestyle factors, including sedentary behavior. Low or decreasing physical activity levels together with unbalanced and high-energy diets, commonly called Western-style diets, abundant in carbohydrates and saturated fats usually lead to increased visceral fat accumulation [144,272,273]. The high extent to which environmental effects can provoke epigenetic responses in obesity is clearly visible after satisfactory treatment (e.g., bariatric surgery), where the improvement in IS and lowering of the levels of inflammatory markers have been reported [274,275].

Therefore, all therapeutic strategies in IR coexisting with obesity should primarily consider the interruption of the vicious cycle of illness in the system, which includes the following: excess visceral AT, immune response/inflammation, and decreased IS [273].

While lifestyle changes, such as eating a healthy diet (i.e., eating fiber-rich foods that have a low- to medium-glycemic index), exercising regularly, and losing excess weight, can increase IS and decrease IR, not all causes are reversible [273,276]. In severe or class 3 obesity (BMI > 40), lifestyle intervention can be successful short-term; however, there is a high rate of recidivism with individuals returning to or exceeding their previous weight [277]. In such cases, bariatric surgery should be a treatment of choice, also because of its anti-inflammatory effect [274,277]. No medications are specifically approved to treat IR. Yet diabetes medications such as metformin and thiazolidinediones, or TZDs, are insulin sensitizers that lower blood glucose, at least in part, by reducing IR [273].

## Figures and Tables

**Figure 1 ijms-24-09818-f001:**
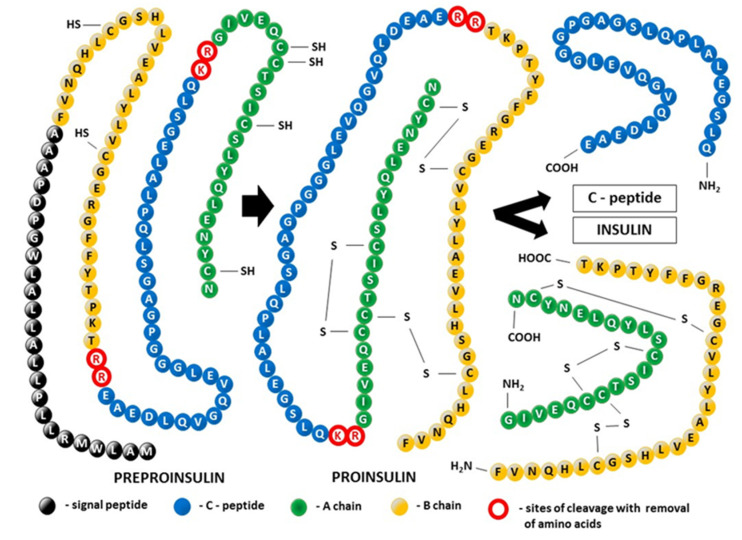
The formation of mature insulin by post-translational modifications to the tertiary structure of the insulin precursors: preproinsulin and proinsulin. Proteolysis of preproinsulin (110 amino acid chain) with the removal of the signal peptide produces proinsulin. Proinsulin has 86 amino acids containing 6 cysteine residues, and the folded protein contains 3 intramolecular disulfide bridges between the A and B chains. C-peptide is part of proinsulin, which is cleaved prior to co-secretion with insulin (51 amino acids) from pancreatic beta cells. The biologically active insulin molecule, therefore, contains less than half of the original translation product.

**Figure 2 ijms-24-09818-f002:**
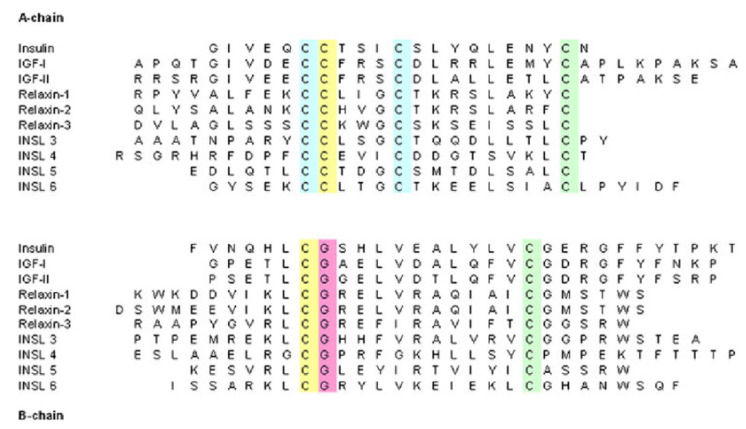
Sequence homology for members within the human insulin–relaxin superfamily of peptide hormones [28]. Conserved cysteines highlighted in blue denote the intra-molecular disulfide bond within the A-chain; yellow and green denote the inter-disulfide bond between the A- and B-chains, respectively. Glycines highlighted in mauve denote conserved residues unique to the B-chain (color figure online) [[28] Nair VB, Samuel CS, Separovic F, Hossain MA, Wade JD. Human relaxin-2: historical perspectives and role in cancer biology. Amino Acids. 2012;43(3):1131-40. doi: 10.1007/s00726-012-1375-y, published with permission of Springer Nature].

**Figure 4 ijms-24-09818-f004:**
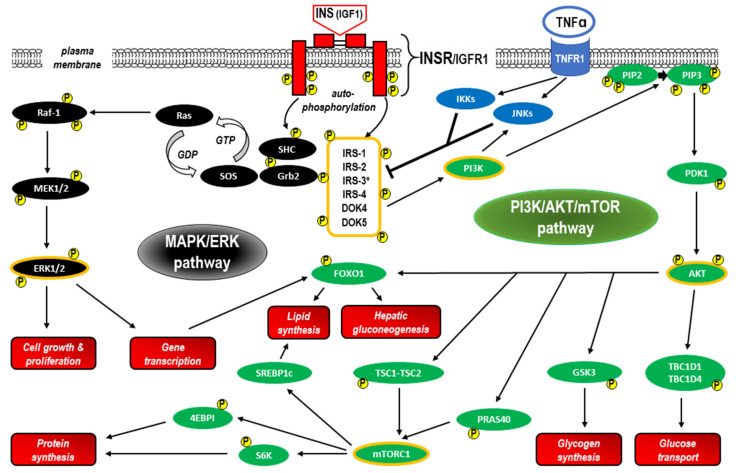
The canonical pathway of insulin receptor (INSR) signaling. The binding of insulin (INS) or IGF-1 to the INSR or insulin-like growth factor receptor 1 (IGFR1) triggers increased tyrosine kinase activity that initiates a cascade of phosphorylation (marked with a yellow circle containing letter P) events that lead to the activation of enzymes that control many aspects of metabolism and growth. Following phosphorylation of adaptor proteins known as insulin receptor substrates (IRSs), the canonical pathway is dichotomized into two major signaling cascades that correspond to the phosphoinositide 3-kinase/protein kinase B/mammalian target of rapamycin (PI3K/AKT/mTOR) and mitogen-activated protein kinase/extracellular signal-regulated kinase (MAPK/ERK) pathways (marked in green and black, respectively). The PI3K/AKT/mTOR pathway is linked exclusively through IRSs and is responsible for most of insulin’s metabolic effects at the cellular level. The MAPK/ERK pathway derives from both IRS and Src homologous and the collagen (SHC) protein and, in addition to the functions common with the PI3K/AKT/mTOR pathway, also regulates gene transcription, cell growth, and proliferation [64]. The diagram has been simplified for clarity. The critical nodes in both pathways (IRSs, ERK1/2, PI3K, AKT, mTORC1) are highlighted with a yellowish-orange border. The main metabolic (anabolic) effects of INSR signaling are listed in red rounded rectangles. Crosstalk from tumor necrosis factor-alpha (TNF-α)/TNF-α receptor 1 (TNFR1) signaling via c-Jun N-terminal kinases (JNKs) and inhibitor of κB (IκB) kinases (IKKs) is indicated (marked in blue) because proinflammatory cytokines (e.g., TNF-α) may produce systemic insulin resistance by the phosphorylation of IRS1. The remaining abbreviations: 4EBPI—4E-binding protein 1, an eukaryotic translation initiation factor; AKT—protein kinase B; DOK4, DOK5—docking protein 5 and 6, also known as IRS-5 and IRS-6, respectively; ERK1/2—extracellular signal-regulated kinases; FOXO1—forkhead homolog in rhabdomyosarcoma 1; GDP, GTP—guanosine diphosphate, guanosine triphosphate; Grb2—growth factor receptor-bound protein 2; GSK3—glycogen synthase kinase-3; MEK1/2—mitogen-activated protein kinase (isoforms 1 and 2); mTORC1—mechanistic (previously referred to as mammalian) target of rapamycin complex 1; PDK1—phosphoinositide-dependent protein kinase 1; PIP2, PIP3—phosphatidylinositol 4,5-bisphosphate, phosphatidylinositol 3,4,5-trisphosphate; PRAS40—proline-rich Akt substrate 40 kDa; Raf-1—(Rapidly accelerated fibrosarcoma) proto-oncogene serine/threonine-specific protein kinase; Ras—(Rat sarcoma virus)GTPase; S6K—protein S6 kinase; SOS—(Son of sevenless), refers to a set of genes encoding guanine nucleotide exchange factors (GEFs); SREBP1c—sterol regulatory element-binding protein 1C; TBC1D1, TBC1D4—RabGTPase-activating proteins (RabGAPs); TSC1-TSC2—tuberous sclerosis complex consisting of the TSC1 (also known as hamartin) and TSC2 (also known as tuberin) proteins. * only in rodent brains and adipocytes.

**Figure 5 ijms-24-09818-f005:**
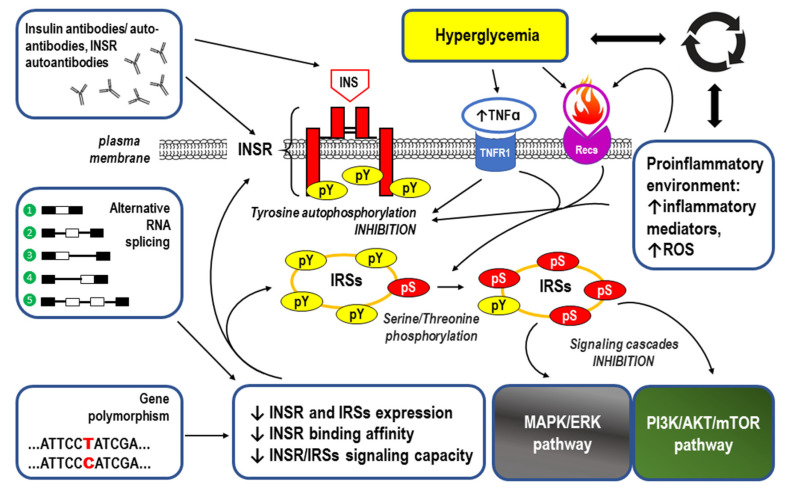
The main underlying causes/mechanisms of insulin resistance (IR) (successive boxes going counter-clockwise from the top-left): Antibodies including autoantibodies antibodies can be produced against both insulin (INS) and the insulin receptor (INSR). Type B IR syndrome is a rare disorder caused by autoantibodies to the insulin receptor [125,134]. Alternative RNA splicing events including ❶—exon retention, ❷—exon skipping, ❸—alternative 5′ donor sites, ❹—alternative 3′ acceptor sites, and ❺—mutually exclusive exons can lead to a splicing mutation that may occur in both introns and exons and disrupt existing splice sites, create new ones, or activate the cryptic ones [135]. Type A IR is caused by a mutation in the INSR gene [136,137]. Defects in the INSR structure and function may be caused by the common type 2 diabetes (T2D) Gly971Arg polymorphism that is responsible for altered tyrosine phosphorylation at a specific site in insulin receptor substrate 1 (IRS-1) [138]. Both alternative RNA splicing and gene polymorphism may significantly affect INSR/IRSs downstream signaling, producing IR-dependent inhibition of the MAPK/ERK and PI3K/AKT/mTOR pathways [122,123,129,139]. A proinflammatory environment and hyperglycemia create a vicious cycle of the complex cause-and-effect relationship, causing IR through inhibition of the tyrosine autophosphorylation (pY) within the INSR and serine/threonine phosphorylation (pS) of the insulin receptor substrates (IRSs). Reactive oxygen species (ROS) and signaling via membrane-bound receptors for tumor necrosis factor receptor 1 (TNFR1) and the receptors for other proinflammatory cytokines (Recs) play crucial roles [122,123,126,140].

## Data Availability

Not applicable.

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
