# Peer review of "Molecular Mechanisms for the Vicious Cycle between Insulin Resistance and the Inflammatory Response in Obesity"

_ijms, 2023, doi:10.3390/ijms24129818_

Round 1
Reviewer 1 Report
Manuscript review response: “Molecular Mechanisms of the Vicious Cycle between Insulin 2 Resistance and the Inflammatory Response in Obesity”.
Manuscript ID: ijms-2421577
Comments
It is an article that could provide relevant information about the relationship between inflammation and insulin resistance in obesity but is necessary there are some changes, add some inscriptions and put some bibliographic citations and others.
Line 39-40 you could add more information about low-grade inflammation or different kinds of inflammation in the body.
Line 51-53 Could you mention which cells in the human body synthesize insulin?
Line 64-68 you could add a Basic Local Alignment Search Tool (BLAST) for protein and gene, so we could make this paragraph more understandable.
Line 75 is necessary to change the title (Posttranslational modification of the tertiary structure of insulin) because in this section had been writing the mechanism of insulin synthesis.
Line 122 you could put these two sections together.
“1.2. Insulin receptor (INSR)” and “1.2.1. INSR structure”
Line 135-138 this part is repeated in previous paragraphs.
Line 144-154 you need to delve into this whole section to make it more understandable.
Line 157-158 What are differentiating signals and is the dominant isoform in physiologic conditions?
Line 158-164 It is convenient to add this section in insulin receptor signalling.
Line 363 Figure 4 describes the role of TNFR1 in insulin resistance and inflammation, however very little is described in the paper about this.
Line 392-394 The methylation of the insulin gene or its receptor, the regulation of miRNAs and the acetalization of chromatin could be included within the epigenetic mechanisms.
Line 856 you have writing “Molecular Mechanisms of the Vicious Cycle between Insulin 2 Resistance and the Inflammatory Response in Obesity” but in your Concluding remarks you mentioned “epigenetic changes” maybe you could rewrite the title or to talk more about Molecular Mechanisms of the Vicious Cycle
Is necessary understand and write molecular mechanisms of insulin resisting and their association between both. You could characterize the types of inflammation that occur in the body and put there the different mechanisms that participate in the inflammation that you are considering in the manuscript
The authors could do a moderate editing of the English language..
Reviewer 2 Report
This is a very well written and detailed review on the molecular mechanisms of IR and inflammatory response in obesity.
A nice addition would be a short paragraph or a visual outline on the different therapeutical strategies and their respective target.
Congratulations to the author for his work!
Round 2
Reviewer 1 Report
It is a manuscript that compiles the role of insulin in adipocytes, I agree with your publication in this journal
Minor editing of English language required